# Analysis of Heavy Metal Contaminants and Mobility in Sewage Sludge-Soil Mixtures for Sustainable Agricultural Practices

**Agata Janaszek and Robert Kowalik ***

Faculty of Environmental, Geomatic and Energy Engineering, Kielce University of Technology,
25-314 Kielce, Poland; ajanaszek@tu.kielce.pl
* Correspondence: rkowalik@tu.kielce.pl

**Abstract:** This study presents a comprehensive analysis of the potential utilization of sewage sludge in agriculture, focusing on the assessment of heavy metal contaminants and their mobility in sewage sludge-soil mixtures. The innovative approach of investigating heavy metal fractions in these mixtures sheds light on their environmental implications. In this study, sludge and soil samples from three different soil categories were collected, and the mobility of heavy metals was investigated using sequential BCR analysis. A thorough assessment of the risk of environmental contamination associated with the agricultural use of sludge was also carried out. This study included the calculation of various risk indicators, such as the Geoaccumulation Index of heavy metals in soil (Igeo), the risk assessment code (RAC), and the author's element mobility ratio (EMR), which included a comparison of the overall metal concentrations in sludge, soil, and mixtures. This study demonstrates that the key to using sludge is to know the form of mobility of the metals present in the sludge and how they behave once they are introduced into the soil.

**Keywords:** sewage sludge; heavy metals; soil; ecological indicators; mobility; sludge-soil mixture; environmental pollution





## 1. Introduction

Unceasing industrial growth and the expansion of infrastructure have triggered the ongoing deterioration of our environment. Contaminants, such as industrial by-products and agricultural runoff, contribute to this decline. The resulting harm not only endangers human well-being and survival but also poses a threat to all forms of life on our planet. Aluminum, in particular, displays high reactivity with both oxygen and carbon. Furthermore, it has the capacity to inflict substantial harm on human health and has been associated with the emergence of conditions like autism spectrum disorders, Alzheimer's disease, and neurotoxicity in the central nervous system [1]. Heavy metals are non-degradable and tend to accumulate within living organisms, potentially causing several disorders and illnesses [2].

Wastewater sewage sludge is a by-product of wastewater treatment processes, and as the global population continues to grow and economies develop, the global production of wastewater sludge is steadily increasing [3]. While this does pose challenges, there are also processes available that can be implemented in more confined spaces. This presents a challenge for wastewater sludge management systems, particularly in urban areas where space for disposal is limited [3,4].

Sewage sludge contains high levels of organic compounds and essential nutrients for plant growth, making it a potential resource in agriculture as a natural fertilizer while simultaneously addressing the issue of sludge management [4]. However, wastewater sludge can also contain pathogens, heavy metals, and other toxic substances, which can pose a threat to human health and the environment if not properly managed [3,5]. Toxic metal pollution is caused by the industrial waste discharged into water ecosystems, which

severely threatens humanity and all living creatures on earth [6]. In aquatic environments, non-biodegradable metals are prevalent and exhibit a propensity to accumulate in living organisms, potentially giving rise to disorders and illnesses over time [7].

In accordance with the Municipal Sludge Management Strategy [5], there is an increasing emphasis on promoting the natural utilization of wastewater sludge, supported by economic and environmental considerations [8,9]. For both sewage sludge and soil, adherence to prescribed heavy metal levels is imperative. Nonetheless, like any other method, this approach possesses both opportunities and constraints. The application of municipal wastewater sludge to land is only permissible when it conforms to the criteria outlined in the Minister of Environment Regulation dated 6 February 2015, pertaining to municipal wastewater sludge [10]. These requirements are aimed at ensuring safe use for human health, life, and the environment. The most significant limitation to the agricultural use of sludge is the presence of heavy metals, as well as the content of parasite eggs and pathogens [11]. While the average concentration of trace elements can provide some insight, it is insufficient for a comprehensive assessment of the potential risks associated with natural sludge utilization. This is due to the fact that heavy metals can migrate through soil layers and infiltrate groundwater and surface waters, ultimately contaminating plants and becoming a potential source of harm to humans [11,12]. The mobility of heavy metals largely depends on their chemical composition in the environment and the soil conditions, especially the pH [12,13]. Figure 1 illustrates the practices related to wastewater sludge in various EU countries, while Table 1 presents the limits regulating the possibility of using wastewater sludge for natural purposes.

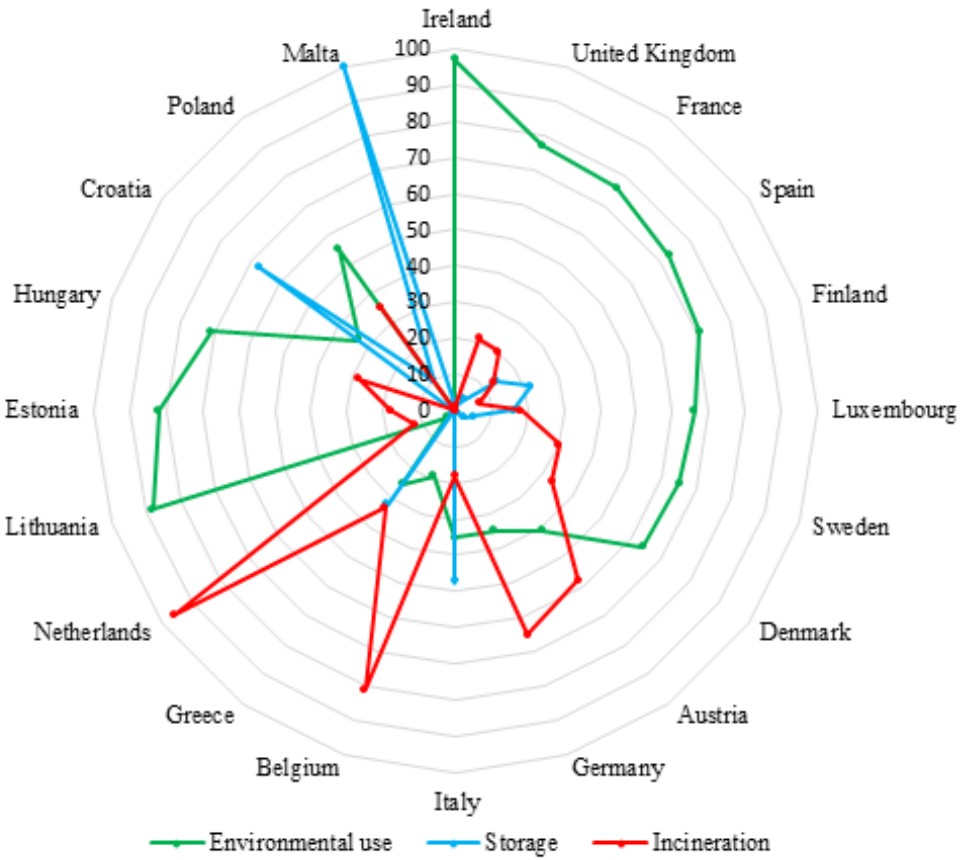

**Figure 1.** Sludge handling in EU countries expressed in %, as of 2022 (authors' chart based on [14]).

**Table 1.** The maximum permissible levels of heavy metals in sewage sludge used for natural purposes, expressed in milligrams per kilogram of dry matter (mg/kg d.m.).

| Heavy Metal | The Allowable Concentrations of Heavy Metals Present in Sewage Sludge Intended for Natural Utilization | | | | | | | | | | |
|---|---|---|---|---|---|---|---|---|---|---|---|
| | Poland [10] | | | EU [15] | Chinese [16] | | United States [17] | South African [18] | Netherlands [19] | Ireland [20] | Malta [21] |
| | In Agriculture and for Reclamation Land for Agricultural Purposes | For the Reclamation of of Land for Non-Agricultural Purposes | When Adapting Land to Specific Needs Arising from Waste Management Plans, Spatial Development Plans | | pH < 6.5 | pH > 6.5 | | | | | |
| Cd | 20 | 25 | 50 | 20–40 | 5 | 20 | 39 | 40 | 1.25 | 20 | 5 |
| Ni | 300 | 400 | 500 | 300–400 | 100 | 200 | 420 | 420 | 30 | 300 | 200 |
| Cr | 500 | 1000 | 2500 | - | 600 | 1000 | - | 1200 | 75 | - | 800 |
| Pb | 750 | 1000 | 1500 | 750–1200 | 300 | 1000 | 300 | 300 | 100 | 750 | 500 |
| Cu | 1000 | 1200 | 2000 | 1000–1750 | 250 | 500 | 1500 | 1500 | 75 | 1000 | 800 |
| Zn | 2500 | 3500 | 5000 | 2500–4000 | 500 | 1000 | 2800 | 2800 | 300 | 2500 | 2000 |

Previous studies have only shown the potential impact of heavy metals from sewage sludge sediments when introduced into the soil. However, there have been no investigations conducted on sludge-soil mixtures as of yet. It is worth noting that such mixtures may contain entirely different concentrations of heavy metals than originally assumed. Therefore, it is essential to monitor the content of heavy metals in sludge-soil mixtures and develop management strategies. These strategies can include suitable methods for removal, recycling, and safe storage. Adopting such an approach can help to mitigate the adverse effects of heavy metals on the natural environment and public health [22].

The aim of this study was to examine the behavior of heavy metals in different types of soil following their introduction in order to understand how their total concentrations change alongside their tendencies for migration. Hence, there is an imperative for a deeper understanding of the chemical compositions of different metals in sewage sludge (SS) and how this speciation changes when SS is introduced into the soil. This understanding is of paramount importance in establishing accurate safety thresholds for metal concentrations in the SS used for agricultural purposes and to enhance the permissible levels of metal accumulation in soil.

This research endeavors to offer a comprehensive overview of the current understanding of metallic contaminants in sewage sludge and their influence on metal behavior within soil. It also aims to introduce novel insights and perspectives regarding the utilization of diverse risk indicators when employing sludge. In particular, we assess how soil contamination stemming from the input of sludge is influenced by the overall metal content and their speciation forms.

This study delves into the ramifications of applying sludge on the metal concentrations and chemical speciation in agricultural soil of varying fertility, ranging from agriculturally rich to less fertile soils. Additionally, we provide recommendations for future research priorities aimed at enhancing the judicious utilization of sewage sludge in agriculture.

## 2. Materials and Methods

Sludge samples were taken from wastewater treatment plants in Sitkowka-Nowiny, located in southeastern Poland. The sewage sludge generated in the wastewater treatment process, both primary and excess, underwent mesophilic fermentation in dedicated fermentation chambers, followed by dewatering at the centrifuge station. Additionally, a portion of the excess sludge (1/3 of the volume) underwent ultrasonic disintegration before being introduced into the fermentation process. The dewatered sewage sludge, along with the fats produced during the wastewater treatment process, were subjected to thermal treatment for safe disposal in the thermal sludge utilization station as part of the completed wastewater treatment plant expansion. The thermal treatment process involves the combustion of the sludge in a fluidized bed incinerator with a hot air chamber equipped with a dry gas cleaning system and a central control system for automation.

### 2.1. Heavy Metal Speciation

The European Community Bureau of Reference (abbreviated as BCR) has developed a four-step procedure that yields the most favorable outcomes for analyzing sewage sludge samples [23,24]. Originally, the classical BCR process consisted of three steps. However, with the introduction of royal water mineralization, this method was modified to allow the separation of heavy metals into the following four distinct fractions: ion-exchangeable (carbonate), reducible (bound to iron and manganese oxides), oxidizable (bound to organic matter), and residual matter [25–27]. The BCR method has found extensive application in numerous studies examining the quality of sewage sludge and heavy-metal-contaminated soils. This technique facilitates the assessment of metal mobility within the tested matrix and the estimation of the potential for contaminant dispersion into the environment [28–30]. Figure 2 illustrates the steps involved in the BCR procedure. For the determination of the total heavy metal content in the soil, aqua regia (a mixture of HCl and $HNO_3$ in a 3:1 ratio) digestion was employed as a method.

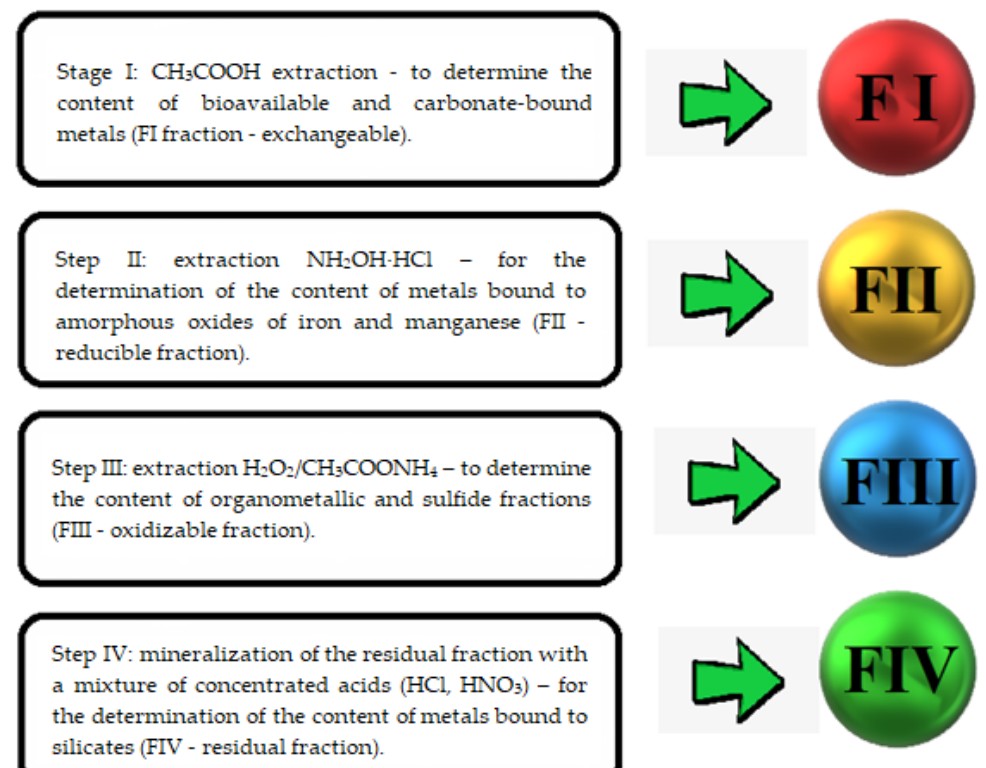

**Figure 2.** Schematic of the sequence of heavy metal speciation analysis using the BCR method (authors' own study).

### 2.2. Soil Samples

Soil samples were taken with an Egner's stick from the Swietokrzyskie region for three different soil types; the locations of the samples are marked in Figure 3, and the soil descriptions are in Table 2. Soil samples of 1 kg were taken from a depth of 20 cm. Mixtures of soil and sludge were then made under laboratory conditions at a ratio of 1:4 (100 g of sludge to 400 g of soil). The samples were mixed after drying by adding the sludge samples to the soil samples and mixing thoroughly under air-dry conditions. After mixing, the samples were kept for 90 days under laboratory conditions. Four independent representative samples were taken from both the soil samples and the sludge-soil mixtures, for which BCR analysis was performed to determine the heavy metal content in total and for all the fractions.

**Table 2.** Description of soil sampling sites.

| Soil Sample | Content of Fraction (%) | | | pH (KCl) | Organic Matter Content | Organic Carbon | Nitrogen Total | C/N Ratio | Absorbable Phosphorus | Soil Sorption Capacity |
|---|---|---|---|---|---|---|---|---|---|---|
| - | fi < 2 μm | 2 < fπ < 50 μm | 50 < fp < 2000 μm | pH | % | % | % | - | mg $P_2O_5$ × 100 $g^{-1}$ | Cmol/kg |
| Soil A | 45 | 40 | 15 | 5.6 | 4.21 | 2.33 | 0.19 | 12.49 | 6.4 | 36.8 |
| Soil B | 30 | 51 | 19 | 5.2 | 3.19 | 1.81 | 0.12 | 15.42 | 5.7 | 11.8 |
| Soil C | 8 | 28 | 64 | 6.7 | 0.78 | 0.56 | 0.06 | 22.27 | 1.2 | 5.3 |

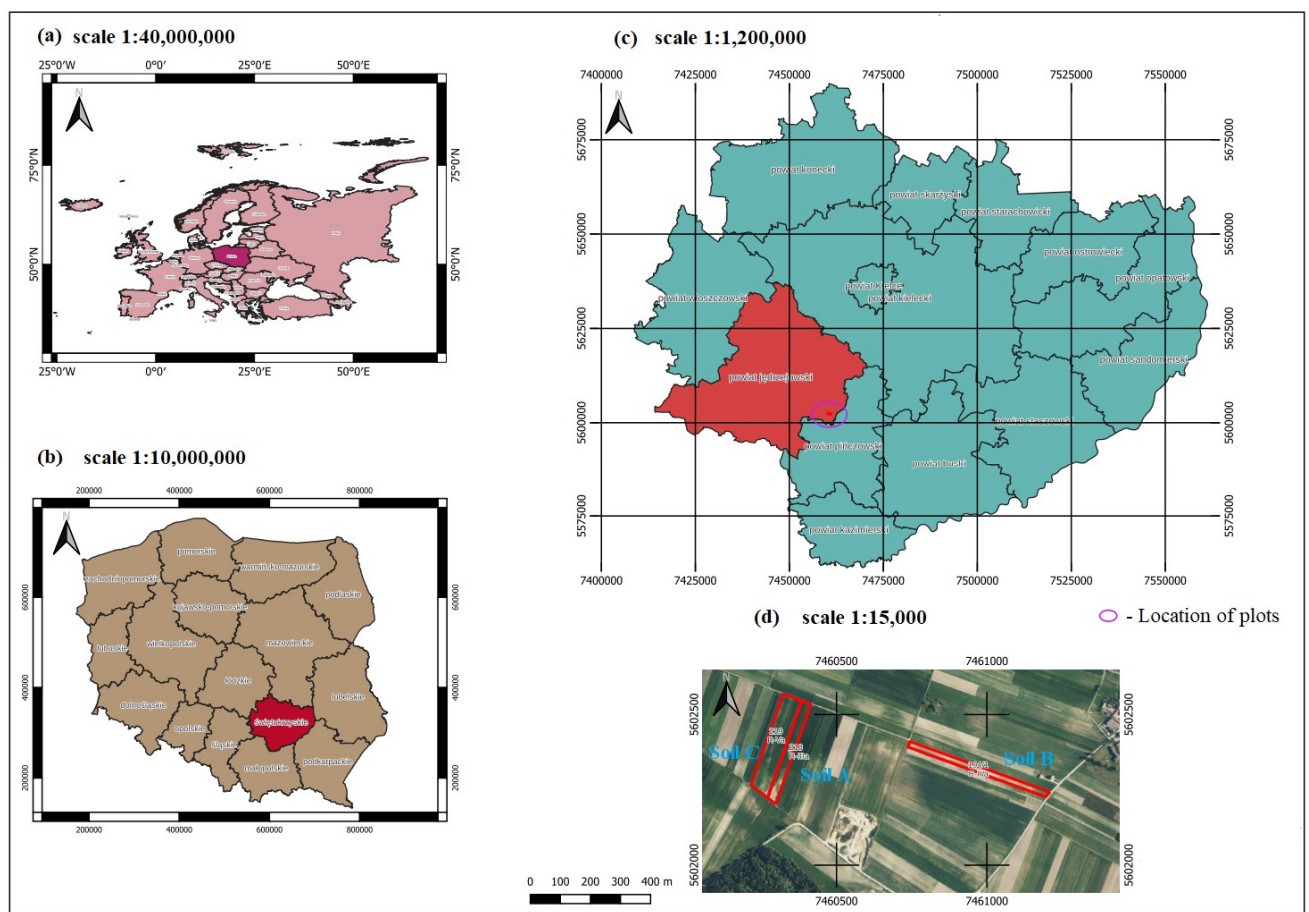

**Figure 3.** Map of soil sampling locations: A, B, and C. (**a**) Map of Europe, (**b**) Map of Poland with indication of the voivodeships, (**c**) Map of districts of Swietokrzyskie voivodeship, (**d**) Orthophoto map with plots 194/1, 218, 219 (authors' own study).

**Soil A**—Class III soils are a category often used in soil classification systems, such as soil taxonomy or the FAO (Food and Agriculture Organization of the United Nations) soil classification system. Class III soils are characterized using specific properties and features that can be described as follows:

Soils of moderate quality: Class III soils are typically of moderate quality. They are not of very high quality but are also not considered poor.

Moderate fertility: Class III soils are usually moderately fertile. This means that they are capable of supporting agricultural crops but may not provide the same level of fertility as Class I or Class II soils.

Agricultural potential: Class III soils are suitable for growing various crops, but they may require additional practices, such as fertilization, to reach their full agricultural potential.

Drainage: The drainage of Class III soils can vary, but they may require some drainage practices to prevent excessive waterlogging and issues related to excess moisture.

Uses: Class III soils are often used for cultivating grains, forage crops, some vegetables, and other crops. They may also be utilized for construction or recreational purposes.

**Soil B**—Class IV soils are characterized using specific properties and features that can be described as follows:

Low soil quality: Class IV soils are generally of low quality. They are not considered to be soils of high agricultural quality.

Low fertility: Class IV soils are nutrient-poor and typically exhibit low fertility, meaning that they are not suitable for the cultivation of many types of crops.

Limited agricultural use: Class IV soils may only be suitable for the cultivation of a few plant species, and even then, they may require a significant input of fertilization and other soil quality improvement practices.

Drainage issues: Class IV soils often have drainage issues, which can lead to excess water retention and the development of overly wet areas.

Alternative uses: Due to their low quality and limited agricultural potential, Class IV soils may be more suitable for other purposes, such as recreation, construction, or environmental preservation.

**Soil C**—Class V soils are characterized by specific properties and features that are described as follows:

Very low soil quality: Class V soils are soils of very low quality. They are considered unsuitable for the cultivation of most agricultural crops.

Minimal fertility: Class V soils are typically nutrient-poor and have minimal capability to support any form of agricultural cultivation.

Limited agricultural use: These soils are usually too poor and unsuitable for traditional agricultural cultivation. Only very limited agricultural use may be possible with intensive fertilizer application and other soil quality improvement practices.

Serious drainage issues: Class V soils often have serious drainage problems, leading to prolonged waterlogging and the creation of excessively wet environments.

Alternative uses: Due to their low quality and inability to support traditional agriculture, Class V soils may be more suitable for other purposes, such as environmental conservation, recreation, or natural preservation [31–33].

### 2.3. Heavy Metal Accumulation Risk Indicators

When the heavy metal content is compared with the limit values reported in the literature, it is only possible to approximate the probability of contamination, which does not provide comprehensive information on soil quality. The key to effectively assessing the heavy metal contamination of soil is the use of pollution indicators. Some of the first indices were created by Muller [34] and Hakanson [35]. Pollution indexes are now regarded as a tool for the geochemical assessment of the soil environment. In addition, contaminant indices are important for monitoring soil quality, especially in agroecosystems.

2.3.1. Geoaccumulation Index of Heavy Metal in Soil (Igeo)

This approach, pioneered by Muller [34], provides a means to ascertain and categorize the degree of pollution in sediments and soil across a spectrum of five levels, ranging from uncontaminated to highly contaminated. The Geoaccumulation Index (Igeo) is primarily contingent on the concentration of heavy metals within the soil substrate at the point where the pollutant is potentially introduced. The formula for Igeo is defined as per the equation below [34,36]:

$$\text{Igeo} = log_2 \frac{C_n}{1.5 \cdot B_n} \tag{1}$$

where

$C_n$—the concentration of individual *HM*s (mg/kg d.m).

$B_n$—the geochemical background value (mg/kg d.m).

The total heavy metal content for the individual soils studied, obtained using the BCR method, was used as the geochemical background value for the formula.

### 2.3.2. Risk Assessment Code (RAC)

By considering both the total metal concentration and chemical speciation, the risk assessment code (RAC) presents a quantitative approach to assessing the bioavailability of heavy metals. This metric was first introduced by Perin and colleagues in 1985 [37], with the assumption that the fraction *FI*, which is associated with carbonates, exhibits the highest mobility and, thus, poses the greatest risk of heavy metal contamination in the soil. However, it is important to note that this indicator only partially represents the true risk of metal migration in the soil matrix, as heavy metals present in the reducible and oxidizable fractions also tend to migrate within the soil medium. The calculation of the risk assessment code is based on the following formula:

$$\text{RAC} = \frac{FI}{HM} \cdot 100\% \tag{2}$$

where

*FI*—the metals' content in fraction *FI* (mg/kg d.m).
*HM*—the overall concentration of heavy metals (mg/kg d.m).

### 2.3.3. Element Mobility Ratio (EMR)

Considering the mobility of heavy metals, it becomes evident that only fraction *FIV* remains completely immobile within the soil. Conversely, fractions *FI* and *FII* exhibit the highest mobility, while fraction *FIII* may acquire mobility under specific conditions. An example of this would be when the soil organic matter is decomposed by microorganisms or fungi or when rainwater contains the dissolved ozone resulting from lightning strikes. The EMR (element mobility ratio) assesses the heavy metal group element content based on their presence in four distinct fractions, with each fraction assigned an appropriate weighting. In this study, the authors propose an index to comprehensively address the issue of heavy metal mobility in risk analysis. This index is represented by the following formula:

$$\text{EMR} = \frac{FI + 0.7 \cdot FII + 0.3 \cdot FIII}{\sum FI \div FIV} \tag{3}$$

Limits for all indicators and risk classifications are summarised in Table 3.

**Table 3.** Classification of indicators [28,34].

| Igeo | RAC | EMR | Risk Value |
|------|-----|-----|------------|
| Igeo ≤ 0 | <0.01 | 0 < EMR ≤ 0.1 | Lack of pollution |
| 0 < Igeo ≤ 1 | 0.01 ÷ 0.1 | 0.1 < EMR ≤ 0.3 | Low pollution |
| 1 < Igeo ≤ 3 | 0.11 ÷ 0.3 | 0.3 < EMR ≤ 0.4 | Average pollution |
| 3 < Igeo ≤ 5 | 0.31 ÷ 0.5 | 0.4 < EMR ≤ 0.5 | High pollution |
| 5 < Igeo | >0.5 | 0.5 < EMR | Extreme pollution |

## 3. Results

Figures 4–6 show the total heavy metal content of the sampled sludge, soil, and sludge-soil mixture. The highest concentrations were recorded for lead and copper. The metal content in the soil samples was significantly lower than in the sludge samples. It is worth noting that the sludge-soil mixture showed similar levels of metal concentrations to those observed in the soil samples alone. This is extremely important in the sense of using sludge for fertilizer purposes. It should be noted that the applicable standards for metal concentrations vary between countries; in Poland, the permissible concentration of copper is 1000 mg/kg per dry weight, while the Netherlands has the most stringent requirement of only 75 mg/kg per dry weight. It is worth noting that the metal concentrations in the sludge itself did not meet Dutch standards, but the sludge-soil mixtures did.

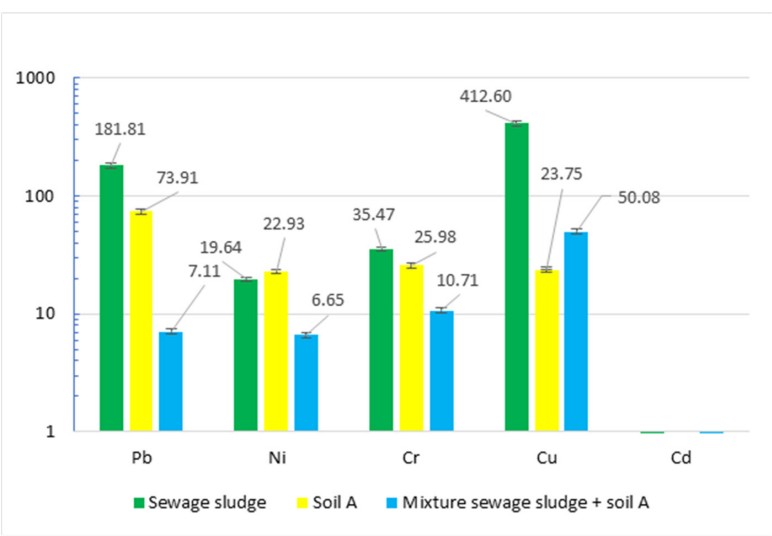

**Figure 4.** Summarized heavy metal content of sewage sludge, soil A, and sludge-soil mixture for soil A.

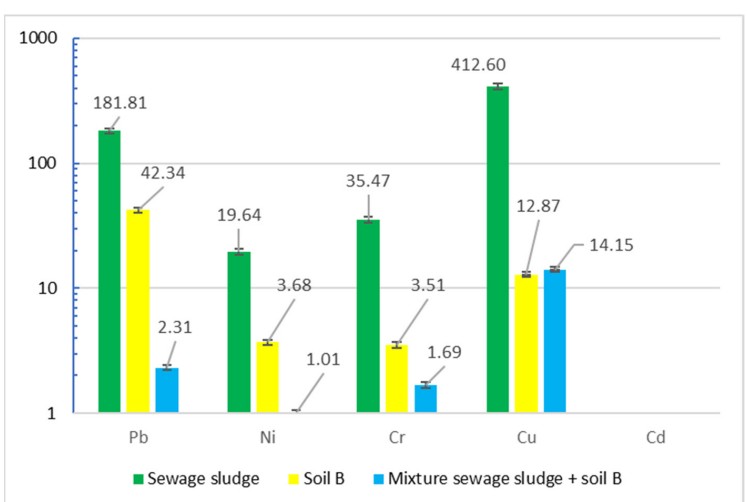

**Figure 5.** Summarized heavy metal content of sewage sludge, soil B, and sludge-soil mixture for soil B.

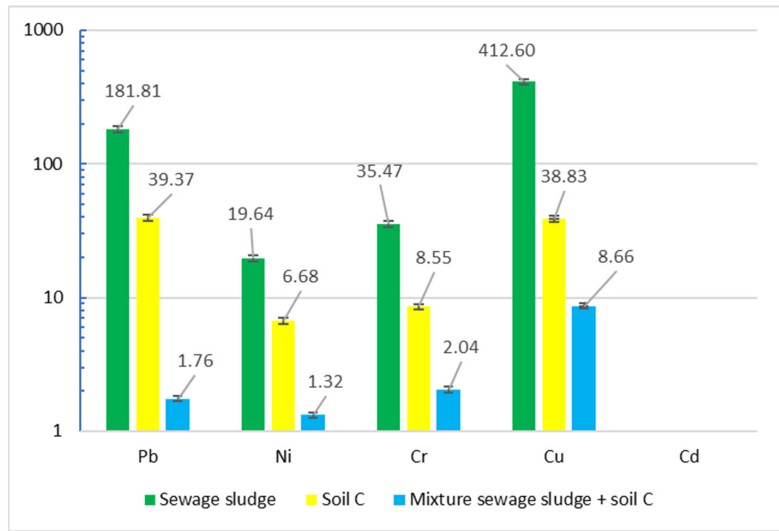

**Figure 6.** Summarized heavy metal content of sewage sludge, soil C, and sludge-soil mixture for soil C.

### 3.1. Geoaccumulation Index of Heavy Metal in Soil (Igeo)

For the sludge and the three soils tested, the Igeo value was recalculated before mixing the samples. The results were compiled by taking the Cn metal content as the value in the sludge, while Bn was used as the heavy metal content of individual soils. The results are shown in Figure 7.

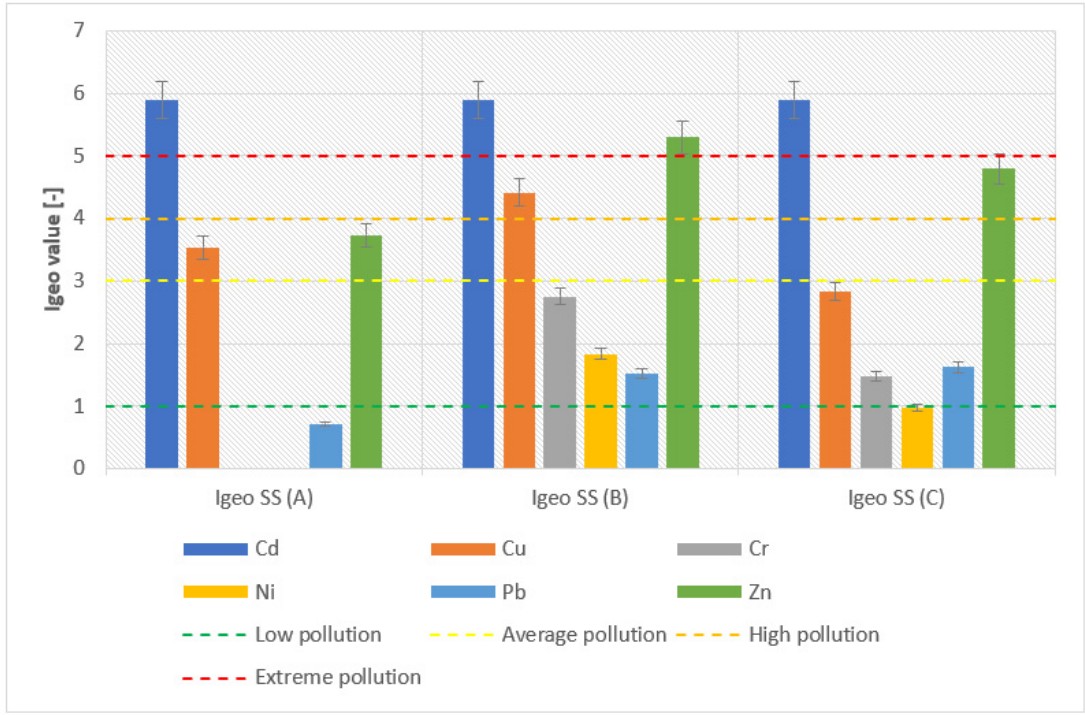

**Figure 7.** Igeo indicator of heavy metals in sewage sludge and soil.

As can be seen, cadmium and zinc for all three cases of sludge introduction into the soil showed a high risk of contamination. Copper showed a high risk for mixtures with soil A and soil B. The other metals showed a low or medium risk of contamination.

### 3.2. Risk Assessment Code (RAC)

Figure 8 presents the results of the RAC index for sludge, soil, and soil/sludge mixture samples. A high risk of contamination, according to the RAC indicator, only occurred for cadmium in the sludge samples. In the soil samples, all metals showed low levels of risk of contamination. However, in the mixed soil and sludge samples, a moderate risk level was observed for cadmium (soil A), as well as for zinc and copper in all three soil samples. Nickel and lead in sludge-soil mixture B also showed a moderate risk of contamination.

### 3.3. Element Mobility Ratio (EMR)

EMR is an indicator that only takes into account the mobility of heavy metals as they are found, i.e., their tendency to migrate into the ground. As can be seen, the heavy metals found in the soil had a higher mobility (Figure 9). Copper and chromium in soil C reached the extreme risk value due to migration risk. The other heavy metals in the soil showed a medium to high risk of migration. In sludge, on the other hand, all metals except nickel and cadmium showed a low risk of migration. Sludge-soil mixtures, nevertheless, showed a lower tendency to migrate metals compared to the soil. This is valuable information, as the heavy metals introduced into soil in non-mobile combinations are more likely to stabilize the metals in soil.

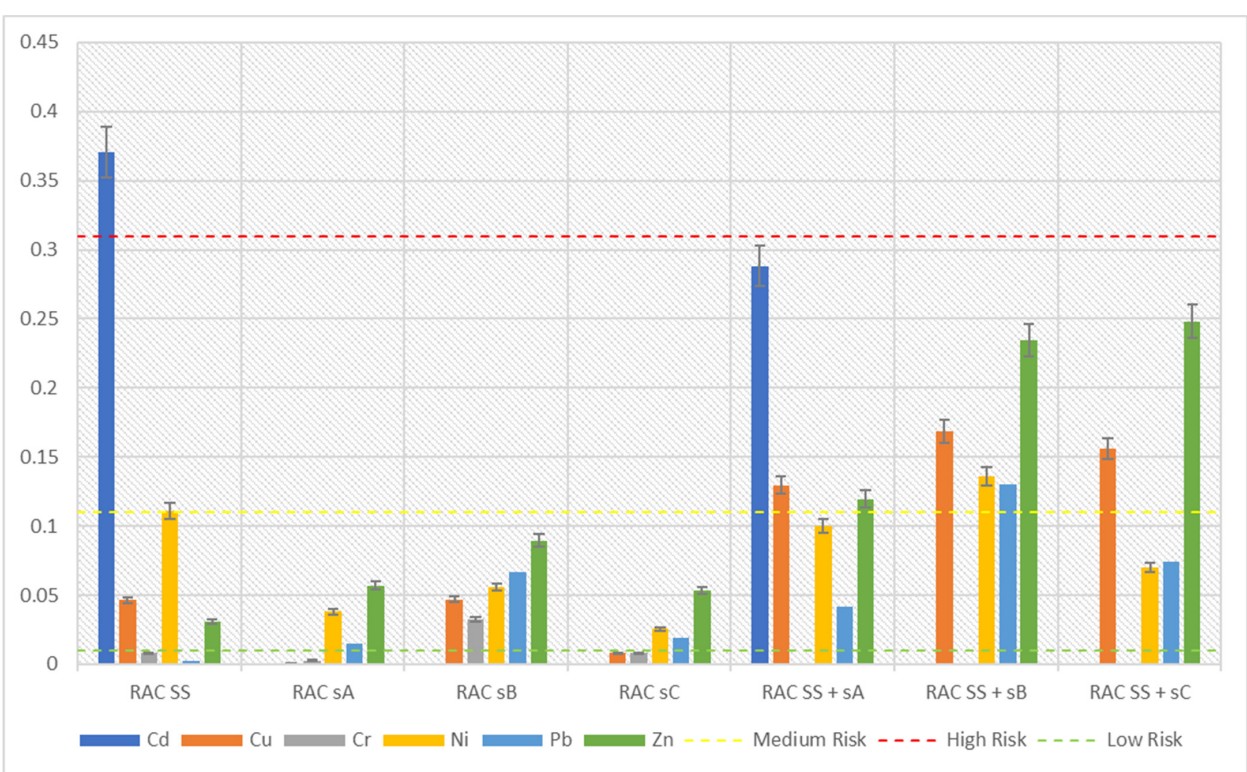

**Figure 8.** RAC indicator of heavy metals in sewage sludge and soil (SS—sewage sludge, sA—soil A, sB—soil B, sC—soil C).

Sludge is an inherent by-product of wastewater treatment. On the one hand, it is a valuable raw material that can be used for agricultural or nature-related purposes; however, for this to happen, the sludge injected into the soil must not pollute it. The requirements placed on sludge are related to its heavy metal and parasite egg content. During wastewater treatment processes, heavy metals are released from the wastewater and then accumulate in the sludge. However, information on the total metal content is not sufficient to draw conclusions about the actual risks associated with their use. The toxicity of heavy metals depends on their speciation form. They can exist in four different forms of mobility, depending on their tendency to migrate deep into the geochemical substrate.

It is estimated that the value of the fertilizing components in sludge generated over one year is about PLN 70 million. Burning the entire annual production of sewage sludge, on the other hand, would involve $CO_2$ emissions to the amount of 250,000 tons [38]. At present, global carbon dioxide emissions have reached record levels and represent one of the most serious global environmental challenges. For this reason, if there is an opportunity to reduce carbon dioxide emissions by reducing the amount of sludge incinerated while recovering a valuable raw material, it is vital that this potential is fully exploited [39].

Sludge contains compounds that are similar to natural humus compounds, which offers the hope of increasing humus levels in soil through its utilization. This approach may contribute to solving the problem of low humus levels in Polish soils.

Heavy metals are found in sludge as well as in soil. The research carried out was aimed at simulating the behavior of heavy metals when they are introduced into soil. Heavy metals in sludge-soil mixtures showed similar concentration levels to soil matrices but a lower mobility.

The differences in the potential agricultural use of sewage sludge in different countries arise from various factors, such as legal regulations and environmental standards, as well as specific climatic and geological conditions. The Netherlands and Malta serve as examples of countries where regulations regarding sewage sludge applications in agriculture are particularly stringent.

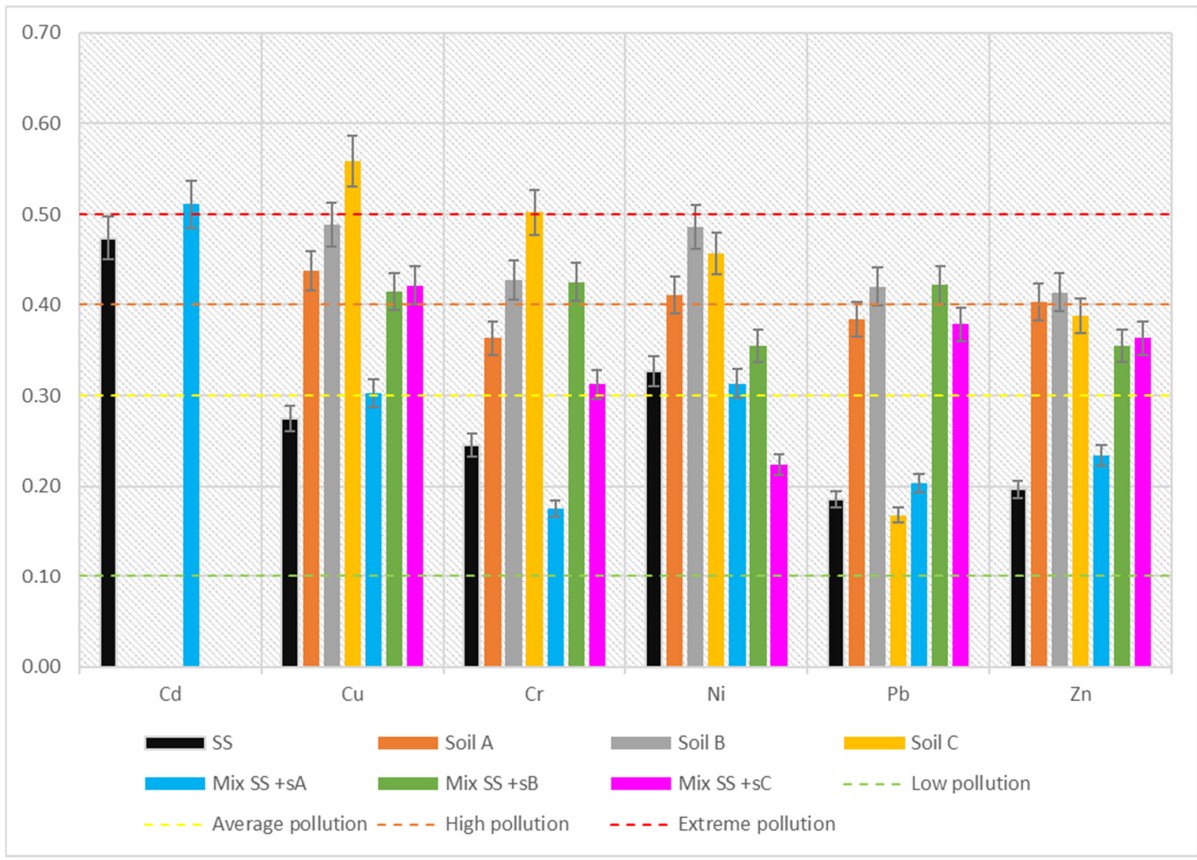

**Figure 9.** EMR indicator of heavy metals in sewage sludge and soil (SS—sewage sludge, sA—soil A, sB—soil B, sC—soil C).

Highly populated areas: The Netherlands and Malta are small countries with relatively high population densities. As a result, a significant amount of sewage sludge is generated within a limited area. Stringent regulations concerning sewage sludge application in agriculture often stem from the necessity to control and mitigate the impact of these sludges on the environment and public health.

Legal norms regarding the quantity of metals in sewage sludge are frequently established to ensure consistency and facilitate waste management. Determining the maximum permissible levels of metals in sewage sludge is more measurable and controllable than monitoring the mobility of metals in the environment. This approach is simpler, although it does not fully reflect the actual risk of metal migration. Introducing mobility criteria into legal standards is significantly more complex, as it requires accounting for the various factors influencing this mobility.

Measuring the mobility of metals in sewage sludge and the natural environment is costly and technically challenging. Therefore, specifying the quantities of metals in sewage sludge is more practical and accessible from the perspective of enforcing legal standards. It is advisable to introduce appropriate indicators and models to minimize the number of required tests.

## 4. Conclusions

In the research conducted in this study, a comprehensive analysis of the potential agricultural utilization of sewage sludge was carried out. The examined samples of sludge, soil, and sludge-soil mixtures met the legal requirements for the heavy metal content in most countries, except for the Netherlands. The total content of heavy metals was significantly reduced in the sludge-soil mixtures, particularly for lead and copper, which displayed the highest concentrations in the sludge. In the case of other metals, the reduction

was less pronounced. The Geoaccumulation Index (Igeo), which only assessed the total metal content in sludge and soil while simulating the contamination risk, indicated a high risk for cadmium in all soil types, zinc in soils B and C, and copper in soil B. The remaining metals exhibited moderate or low risks. The risk assessment code (RAC), focusing on mobility, only identified a high risk for cadmium in the sewage sludge sample, while heavy metals in sludge-soil mixtures mostly demonstrated moderate risks. For the authors' proposed EMR (elemental mobility ratio) index, no heavy metal in sludge-soil mixtures reached an extreme pollution risk level. High risks for sludge-soil mixtures were observed for copper in soils B and C, chromium (soil C), and lead (soil B).

In summary,

1. This research indicates that the total concentration of heavy metals is not a reliable indicator to assess the feasibility of using sewage sludge for agricultural purposes.
2. Heavy metals in a more mobile form, when introduced into the soil under dry conditions, transition into less mobile forms, irrespective of their soil class.
3. In sewage sludge, all metals except nickel and cadmium exhibited a low migration risk. Conversely, sludge-soil mixtures showed a reduced tendency for metal migration compared to soil alone. This observation is valuable as it suggests that heavy metals introduced into the soil in the form of mixtures are less mobile and are likely to be stabilized in the soil.
4. Soil quality does not have a significant impact on the stabilization of heavy metals. However, it is crucial to introduce microorganisms into the soil and expose sludge-soil mixtures to atmospheric conditions, and will be continued in subsequent research.

**Author Contributions:** Conceptualization, A.J.; methodology, R.K.; software, A.J.; validation, R.K.; formal analysis, R.K.; investigation, R.K. and A.J.; resources, A.J.; data curation, R.K.; writing—original draft preparation, R.K. and A.J.; visualization, R.K. and A.J.; project administration, A.J.; funding acquisition, A.J. All authors have read and agreed to the published version of the manuscript.

**Funding:** This project is supported by the program of the Minister of Science and Higher Education under the name: "Regional Initiative of Excellence" in 2019–2023 project number 025/RID/2018/19 financing amount PLN 12,000,000.

**Data Availability Statement:** The datasets supporting the results of this article are included within the article and its additional files.

**Conflicts of Interest:** The authors declare no conflict of interest.

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
