# Peer review of "Analysis of Heavy Metal Contaminants and Mobility in Sewage sludge-soil Mixtures for Sustainable Agricultural Practices"

_water, doi:10.3390/w15223992_

Round 1
Reviewer 1 Report
Comments and Suggestions for Authors
The author methodically applies the European Community Bureau of Reference (BCR) sequential extraction procedure to characterize the speciation of heavy metals in sludge samples. This paper also employs the Igeo and the Risk Assessment Code (RAC) to quantitatively assess the bioavailability and migration risk of heavy metals in soils. Overall, the manuscript provides a comprehensive study on the environmental risks associated with heavy metal content in sewage sludge from wastewater treatment plants. I believe such manuscript is worth published, however, some problem exists:
1. The use of Igeo and RAC is appropriate; however, the manuscript should address the inherent uncertainties in these models. The author should also consider whether these indices adequately capture the long-term and ecological risks associated with heavy metal migration.
2. The manuscript could also benefit from a very simple discussion on the policy implications at both national and international levels.
3. The author should address the extent to which the findings can be generalized beyond the specific contexts studied. Are the results applicable to other regions with different soil types, climates, and agricultural practices? As the samples for experiments were taken from certain place. However, is it representative enough for a solid and general conclusion?
4. Also, more details about the sample should be included. It should also discuss the sampling strategy, including the number of samples, their geographic distribution, and the temporal aspects of sampling.
5. Figure 1 need to state the copyright. Seems it come from other literature.
6. It is doubt that “However, there have been no investigations conducted on soil-sediment mixtures as of yet.” For example, https://doi.org/10.1016/j.teac.2021.e00125
Author Response
- Figure 1 need to state the copyright
Thank you for your contribution, this is our original drawing, inserted in the work.
- Conclusions part can be more summarised.
Thank you for your attention, corrected
- The samples for experiments were taken from certain place. However, is it representative enough for a solid and general conclusion?
Thank you for your attention. Sludge samples were washed from the sewage treatment plant, the test was repeated four times to remove coarse errors. The results considered are those averaged over the sludge samples. The same procedure was carried out for the soil samples. Four sites were sampled over an area of one hectare, for each of the soils analysed. The results were averaged.
- The use of Igeo and RAC is appropriate; however, the manuscript should address the inherent uncertainties in these models. The author should also consider whether these indices adequately capture the long-term and ecological risks associated with heavy metal migration.
Thank you for your comment.
The indices are used in many studies, however, they have their drawbacks, the Igeo index does not account for heavy metal mobility, as we have pointed out. The RAC indicator, on the other hand, only takes into account the most mobile FI fraction ignoring the other conditionally mobile fractions. Our proprietary index, on the other hand, takes into account all fractions in their respective weights, however, it will be further improved in future studies.
- The manuscript could also benefit from a very simple discussion on the policy implications at both national and international levels.
Thank you very much the information has been inserted in the text.
Differences in the potential agricultural use of sewage sludge among different countries arise from various factors, such as legal regulations, environmental standards, as well as specific climatic and geological conditions. The Netherlands and Malta serve as examples of countries where regulations regarding sewage sludge application in agriculture are particularly stringent.
Highly populated areas: The Netherlands and Malta are small countries with relatively high population densities. As a result, a significant amount of sewage sludge is generated within a limited area. Stringent regulations concerning sewage sludge application in agriculture often stem from the necessity to control and mitigate the impact of these sludges on the environment and public health.
Legal norms regarding the quantity of metals in sewage sludge are frequently established to ensure consistency and facilitate waste management. Determining the maximum permissible levels of metals in sewage sludge is more measurable and controllable than monitoring the mobility of metals in the environment. This approach is simpler, although it does not fully reflect the actual risk of metal migration. Introducing mobility criteria into legal standards would be significantly more complex, as it would require accounting for various factors influencing this mobility.
Measuring the mobility of metals in sewage sludge and the natural environment is costly and technically challenging. Therefore, specifying the quantities of metals in sewage sludge is more practical and accessible from the perspective of enforcing legal standards. It would be advisable to introduce appropriate indicators and models to minimize the number of required tests.
- The author should address the extent to which the findings can be generalized beyond the specific contexts studied. Are the results applicable to other regions with different soil types, climates, and agricultural practices? As the samples for experiments were taken from certain place. However, is it representative enough for a solid and general conclusion?
The research was performed in terms of changes in mobility and concentrations of heavy metals in soil-sediment mixtures for different soil types in terms of their agricultural usefulness, from good quality (A) to poor quality (C) soils. In our next studies, we plan to perform an analysis in terms of the influence of weather conditions.
- Also, more details about the sample should be included. It should also discuss the sampling strategy, including the number of samples, their geographic distribution, and the temporal aspects of sampling.
Thank you for your attention. It has been introduced in point 2.2
- Figure 1 need to state the copyright. Seems it come from other literature.
This is our own chart made from data from the literature provided. information on this topic was introduced, thank you.
- It is doubt that “However, there have been no investigations conducted on soil-sediment mixtures as of yet.” For example, https://doi.org/10.1016/j.teac.2021.e00125
Thank you for your attention, the sentence has been removed.
Reviewer 2 Report
Comments and Suggestions for Authors
The work is interesting and has potential, but needs a lot of work, especially in terms of methodology, data presentation and discussion. Detailed comments are provided below:
Line 16 - please provide full name for RAC and EMR before using abbreviations.
Lines 26-28 - yes it does create challenges, however there are some processes that can be used without large spaces. Please rewrite this sentence.
Lines 31-33 References needed.
Line 38 Polish Minister of the Environment regulation. Please rewrite this sentence as both SS and soil must contain HM at certain levels.
Lines 41-42 - what about pathogens and parasite eggs?
Line 47-48 and soil conditions, especially pH.
Line 52 2022 not 202.
Table 1. Please give the actual data for Poland in the form of a range from the Regulation.
Lines 56-57 This is not true. There are many studies on the impact of sewage sludge on soil, e.g. https://doi.org/10.1186/s40538-018-0122-3, https://doi.org/10.1016/j.ecoenv.2021.112070, https://doi.org/10.1016/j.emcon.2022.100200, https://doi.org/10.3390/su13042317
Line 67 Please state the name of the WW treatment plant.
Line 77-78 Move this sentence to the soil subsection.
Line 91-92 In what proportion?
Line 97-98 From what depth? How many replicates? What was the final mass of the samples? How was HM determined, step by step?
Lines 101-144 Soils are very nicely described, but necessary information is missing: what was the particle size composition, how much organic matter was there, what was the pH of the soil samples tested?
Please explain how SS and soil were mixed, in what proportions? Was it dried or wet? How were the samples incubated? Under what conditions?
Line 163 Please add references to the geochemical background.
Line 175 Fraction FI
Line 178 Fraction FIV
Lines 189-191 Please give translation into English
Line 194 Please give any statistical analysis
Figures 4-6 repeat the data from Table 6. Please choose whether to show the data on the figures or in the table.
Line 218 Figure 7? It would be more legible to provide lines for treshold values on the figure.
Line 231 Figure 8? It would be more legible to provide lines for treshold values on the figure.
Line 245 Figure 9? It would be more legible to provide lines for treshold values on the figure.
Lines 247-270 Please provide a discussion based on the results. Now it is based on a reference with the main idea regarding thermal processing and financial value of fertiliser, has not much in common with the topic of the work and the results presented.
Lines 271-296 Please present your conclusions based on the data obtained and do not describe again what has been calculated.
Author Response
Line 16 - please provide full name for RAC and EMR before using abbreviations.
Thank you for your attention, corrected
Lines 26-28 - yes it does create challenges, however there are some processes that can be used without large spaces. Please rewrite this sentence.
Thank you for your attention, corrected
Lines 31-33 References needed.
Thank you for your attention, corrected
Line 38 Polish Minister of the Environment regulation. Please rewrite this sentence as both SS and soil must contain HM at certain levels.
Thank you for your attention, corrected
Lines 41-42 - what about pathogens and parasite eggs?
Thank you for your attention, corrected
Line 47-48 and soil conditions, especially pH.
Thank you for your attention, added
Line 52 2022 not 202.
Thank you for your attention, corrected
Table 1. Please give the actual data for Poland in the form of a range from the Regulation.
Thank you for your attention, added
Lines 56-57 This is not true. There are many studies on the impact of sewage sludge on soil, e.g. https://doi.org/10.1186/s40538-018-0122-3, https://doi.org/10.1016/j.ecoenv.2021.112070, https://doi.org/10.1016/j.emcon.2022.100200, https://doi.org/10.3390/su13042317
Thank you for your attention, the sentence has been removed.
Line 67 Please state the name of the WW treatment plant.
Thank you for your attention, added
Line 77-78 Move this sentence to the soil subsection.
Thank you for your attention, moved.
Line 91-92 In what proportion?
Thank you for your attention, added information 3:1 ratio
Line 97-98 From what depth? How many replicates? What was the final mass of the samples? How was HM determined, step by step?
Thank you for your attention, included information in the text.
Soil samples of 1 kg were taken from a depth of 20 cm. Mixtures of soil and sludge were then made under laboratory conditions at a ratio of 1:4 (100 grams of sludge to 400 grams of soil). Four independent representative samples were taken from both the soil samples and the soil-sediment mixtures, for which BCR analysis was performed to determine the heavy metal content in total and for all fractions.
Lines 101-144 Soils are very nicely described, but necessary information is missing: what was the particle size composition, how much organic matter was there, what was the pH of the soil samples tested?
Thank you, we have added the information in Table 2
Please explain how SS and soil were mixed, in what proportions? Was it dried or wet? How were the samples incubated? Under what conditions?
Thank you for your comment. Information has been added in point 2.2
Line 163 Please add references to the geochemical background.
The total content for heavy metals for the individual soils studied, obtained using the BCR method, was used as the geochemical background value for the formula. Introduced in the text, thank you.
Line 175 Fraction FI
Thank you for your attention, corrected
Line 178 Fraction FIV
Thank you for your attention, corrected
Lines 189-191 Please give translation into English
Thank you for your attention. We apologise for the oversight, the excerpt has been translated.
Line 194 Please give any statistical analysis
Thank you for your attention. The table has been removed in order not to duplicate the information contained in the graphs. We felt that the graphs introduced more meaningful information for the article.
Figures 4-6 repeat the data from Table 6. Please choose whether to show the data on the figures or in the table.
Thank you for your attention. The table has been removed in order not to duplicate the information contained in the graphs. We felt that the graphs introduced more meaningful information for the article.
Line 218 Figure 7? It would be more legible to provide lines for treshold values on the figure.
Thank you for your attention, corrections have been made
Line 231 Figure 8? It would be more legible to provide lines for treshold values on the figure.
Thank you for your attention, corrections have been made
Line 245 Figure 9? It would be more legible to provide lines for treshold values on the figure.
Thank you for your attention, corrections have been made
Lines 247-270 Please provide a discussion based on the results. Now it is based on a reference with the main idea regarding thermal processing and financial value of fertiliser, has not much in common with the topic of the work and the results presented.
Thank you for your attention, corrections have been made
Lines 271-296 Please present your conclusions based on the data obtained and do not describe again what has been calculated.
Thank you for your attention, corrections have been made
Reviewer 3 Report
Comments and Suggestions for Authors
Manuscript ID: water-2709392
Assessment of Heavy Metal Contaminants and Mobility in Sewage Sludge-Soil Mixtures for Sustainable Agricultural Practices
Water
Dear Editor-in-Chief;
The mentioned manuscripts subjects to corrections.
1. The title is to be: Analysis of Heavy Metal Contaminants and Mobility in Sewage Sludge-Soil Mixtures for Sustainable Agricultural Practices
2. The English language should be polished
3. Section 1. To improve the manuscript quality for water treatment, cite the followings; Desalination and Water Treatment 214 (2021) 440-451, Chinese Journal of Chemical Engineering 32 (2021) 472–484, polymers 12 (2020) 1305-1322, processes 7 (2019) 249-268
4. Section 1. Highlight the objective of the work
5. Section 2 needs more scientific elaborations
6. Fig. 3 should be cited
7. Figs 4-7 needs further interpretations
8. Tables 1 & 6. Where are between them?
9. Table 6 is repeated
10. Combine sections 3 & 4
11. Conclusions should reflect abstract
12. 18 of 35 refs is out of date (more than 10 years). Update!
Comments on the Quality of English Language
Manuscript ID: water-2709392
Assessment of Heavy Metal Contaminants and Mobility in Sewage Sludge-Soil Mixtures for Sustainable Agricultural Practices
Water
Dear Editor-in-Chief;
The mentioned manuscripts subjects to corrections.
1. The title is to be: Analysis of Heavy Metal Contaminants and Mobility in Sewage Sludge-Soil Mixtures for Sustainable Agricultural Practices
2. The English language should be polished
3. Section 1. To improve the manuscript quality for water treatment, cite the followings; Desalination and Water Treatment 214 (2021) 440-451, Chinese Journal of Chemical Engineering 32 (2021) 472–484, polymers 12 (2020) 1305-1322, processes 7 (2019) 249-268
4. Section 1. Highlight the objective of the work
5. Section 2 needs more scientific elaborations
6. Fig. 3 should be cited
7. Figs 4-7 needs further interpretations
8. Tables 1 & 6. Where are between them?
9. Table 6 is repeated
10. Combine sections 3 & 4
11. Conclusions should reflect abstract
12. 18 of 35 refs is out of date (more than 10 years). Update!
Author Response
- The title is to be: Analysis of Heavy Metal Contaminants and Mobility in Sewage Sludge-Soil Mixtures for Sustainable Agricultural Practices
Thank you for your attention, corrected
- The English language should be polished
Thank you for your attention, corrected
- Section 1. To improve the manuscript quality for water treatment, cite the followings; Desalination and Water Treatment 214 (2021) 440-451, Chinese Journal of Chemical Engineering 32 (2021) 472–484, polymers 12 (2020) 1305-1322, processes 7 (2019) 249-268
Thank you for your attention, we have enriched the paper, the information contained in the articles. have been cited.
- Section 1. Highlight the objective of the work
Thank you for your attention, corrections have been made
- Section 2 needs more scientific elaborations
Thank you for your attention, corrections have been made
- Fig. 3 should be cited
Thank you for your attention, this is our own graph developed from literature data. Information on.
- Figs 4-7 needs further interpretations
Thank you for your attention, corrections have been made
- Tables 1 & 6. Where are between them?
Thank you for your attention, corrected
- Table 6 is repeated
Thank you for your attention. The table has been removed in order not to duplicate the information contained in the graphs. We felt that the graphs introduced more meaningful information for the article.
- Combine sections 3 & 4
Thank you for your attention, corrected
- Conclusions should reflect abstract
Thank you for your attention, corrected
- 18 of 35 refs is out of date (more than 10 years). Update!
Thank you very much. We have updated the literature, however, we cannot change this with legislation that was introduced more than 10 years ago but is still in use.
Round 2
Reviewer 1 Report
Comments and Suggestions for Authors
The author have properly answer the question. Good for publication.
Author Response
Thank you very much for your review and acceptance of the article.
Reviewer 2 Report
Comments and Suggestions for Authors
In this version, the work is a great improvement on the original version. In my opinion it can be published without further changes.
Author Response

(The authors gave the same response as above.)

Reviewer 3 Report
Comments and Suggestions for Authors
OK
Author Response

(The authors gave the same response as above.)
